# COVID-19 Is an Independent Risk Factor for Detrimental Invasive Fungal Disease in Patients on Veno-Venous Extracorporeal Membrane Oxygenation: A Retrospective Study

**DOI:** 10.3390/jof9070751

**Published:** 2023-07-15

**Authors:** Jens Martin Poth, Jens-Christian Schewe, Felix Lehmann, Johannes Weller, Mathias Willem Schmandt, Stefan Kreyer, Stefan Muenster, Christian Putensen, Stefan Felix Ehrentraut

**Affiliations:** 1Department of Anesthesiology and Intensive Care Medicine, University Hospital Bonn, 53127 Bonn, Germany; 2Department of Anesthesiology, Intensive Care Medicine and Pain Therapy, University Hospital Rostock, 18057 Rostock, Germany; 3Department of Neurology, University Hospital Bonn, 53127 Bonn, Germany

**Keywords:** invasive fungal disease (IFD), invasive fungal infection (IFI), extracorporeal membrane oxygenation (ECMO), candidemia, aspergillosis, COVID-19, SARS-CoV2

## Abstract

Invasive fungal disease (IFD) is associated with the mortality of patients on extracorporeal membrane oxygenation (ECMO). Several risk factors for IFD have been identified in patients with or without ECMO. Here, we assessed the relevance of coronavirus disease (COVID-19) for the occurrence of IFD in patients on veno-venous (V-V) ECMO for respiratory failure. In a retrospective analysis of all ECMO cases between January 2013 and December 2022 (2020–2022 for COVID-19 patients), active COVID-19 and the type, timing and duration of IFD were investigated. Demographics, hospital, ICU length of stay (LoS), duration of ECMO, days on invasive mechanical ventilation, prognostic scores (Respiratory ECMO Survival Prediction (RESP) score, Charlson Comorbidity Index (CCI), Therapeutic Intervention Scoring System (TISS)-10, Sequential Organ Failure Assessment (SOFA) score and Simplified Acute Physiology Score (SAPS)-II) and length of survival were assessed. The association of COVID-19 with IFD was investigated using propensity score matching and uni- and multivariable logistic regression analyses. We identified 814 patients supported with ECMO, and 452 patients were included in further analyses. The incidence of IFD was 4.8% and 11.0% in patients without and with COVID-19, respectively. COVID-19 status represented an independent risk factor for IFD (OR 4.30; CI 1.72–10.85; *p*: 0.002; multivariable regression analysis). In patients with COVID-19, 84.6% of IFD was candidemia and 15.4% represented invasive aspergillosis (IA). All of these patients died. In patients on V-V ECMO, we report that COVID-19 is an independent risk factor for IFD, which is associated with a detrimental prognosis. Further studies are needed to investigate strategies of antifungal therapy or prophylaxis in these patients.

## 1. Introduction

In patients on extracorporeal membrane oxygenation (ECMO), invasive fungal disease (IFD) is associated with mortality [1,2]. Several risk factors for IFD have been identified in patients with or without ECMO. These include host factors, such as the receipt of an organ transplant or a hematologic malignancy [1,3], and host-independent factors, such as influenza A infection, which is a risk factor for pulmonary aspergillosis, irrespective of ECMO support [1,4,5,6,7]. Similarly, renal replacement therapy and sepsis were identified as risk factors for candidemia in patients on and off ECMO [1,8,9].

Coronavirus disease (COVID-19) is associated with candidemia and pulmonary aspergillosis in hospitalized patients [10,11,12]. Recent studies have also described a high incidence of aspergillosis and candidemia in patients with COVID-19 (COVID-19^+^ patients) on ECMO [13,14]. We hypothesize that COVID-19 is an independent risk factor for IFD in patients on ECMO, which has not been investigated so far.

Here, we present the results of a single-center, retrospective study on the relevance of COVID-19 for the incidence of IFD in patients on veno-venous (V-V) ECMO for acute respiratory failure. We further analyze the impact of IFD on the survival of COVID-19 patients on ECMO.

## 2. Materials and Methods

The local Ethics Committee approved this analysis, and the need for individual informed consent was waived (Bonn Medical Faculty Ethics Committee #492/20).

We reviewed all ECMO patients (*n* = 814) treated between 1 January 2013 and 30 December 2022 in our institution. Only patients with V-V ECMO were included in the study (Figure 1). We recorded demographical data, known comorbidities indicated by the Charlson Comorbidity Index (CCI), organ failure at the time of ECMO initiation, as indicated by the Sequential Organ Failure Assessment Score (SOFA), and the need for continuous kidney replacement therapy (CKRT). The length of invasive mechanical ventilation (IMV) before ECMO and the length of stay (LoS) in hospital and in the intensive care unit (ICU) were also recorded. For each patient, survival was predicted using the Respiratory Extracorporeal Membrane Oxygenation Survival Prediction (RESP) score [15]. The total duration of ECMO and time on ECMO prior to the first detection of IFD were assessed. In patients without IFD, this was defined as the total time on ECMO. Successful ECMO-weaning, disease severity assessed using the Simplified Acute Physiology Score II (SAPS) [16], nursing workload indicated via the Therapeutic Intervention Scoring System (TISS-10) [17], total days of IMV and length of survival after hospital discharge were evaluated, as described previously [2]. Due to the patients’ sedation, SAPS-II was recorded without using the Glasgow Coma Scale (GCS). For SOFA scoring, the best assumed/last known GCS prior to disease onset was used. Immunosuppression due to oncological malignancies, malignancy-related therapies, solid organ transplantation, HIV, autoimmune diseases and immunosuppressive therapies were also evaluated according to Rilinger et al. [18]. This definition did not include hydrocortisone for sepsis or dexamethasone for COVID-19.

For each patient of the V-V ECMO cohort, microbiological samples and tests performed during the ICU stay were screened. Test procedures were as described previously [2]: Blood cultures (BCs) were routinely taken with every insertion of central or arterial lines. Upon initiation of ECMO, bronchoalveolar lavage fluid (BALF) was collected for culture. BALF was also tested for galactomannan (GM), supplemented with GM testing of blood. Further samples were taken at the discretion of the treating physician.

Cultural findings of candida spp. in tracheal aspirates or bronchoalveolar lavage fluid (BALF), as well as in urine, were considered to represent colonization rather than infection. Swabs from skin, the nasal orifice or mouth were not considered. Other findings of candida were considered as IFD, in agreement with the most recent definition for proven IFD [3,19]. For *aspergillus* spp., cases with positive galactomannan assays with additional, repeatedly positive cultural findings in tracheal aspirates, BALF, or other body fluids and biopsies were considered to be IFD. This definition of invasive aspergillosis (IA) does not meet the criteria for invasive fungal disease according to the current consensus definition [3]. However, for critically ill patients without hematological malignancies, recent studies derived the aforementioned criteria for the diagnosis of IA, acknowledging that predisposing host factors do not have to be present [20,21].

Binary logistic regression analysis was used to determine the odds ratio (OR) for IFD in COVID-19 patients (Figure 1). For validation, COVID-19 patients were matched to COVID-19-negative (henceforth non-COVID-19) patients utilizing propensity score matching (PSM) with the following treatment-independent variables: SOFA score upon admission, age, gender, height and weight. Univariable binary logistic regression analysis was performed in the PSM cohort, with COVID-19 status as the independent variable and IFD as the dependent variable (Figure 1).

In all patients on V-V ECMO, the OR for IFD was also analyzed via multivariable binary logistic regression, using the same variables as for PSM. In addition, we also included total time on IMV and duration of ECMO before IFD as independent variables to control for these potential confounders.

After hospital discharge, patients were actively followed for survival. Between groups, survival was compared using Kaplan–Meier analysis and the log-rank test for trend [22].

Levels of significance were determined using Mann–Whitney U, Kruskal–Wallis or Fisher’s exact test, where appropriate. The distribution of data was analyzed using the Shapiro–Wilk test. All values are presented as median ± interquartile range (IQR) or mean ± standard error (SE), where applicable. A *p*-value < 0.05 was considered to be significant. Data were analyzed using *R* Version 4.2.2.

## 3. Results

### 3.1. Study Population and Patient Characteristics

During the study period, 814 patients were supported with ECMO (Figure 1). Of 452 patients on V-V ECMO, 118 were treated for COVID-19. A total of 334 patients had respiratory failure due to other causes, and of which, 92 were treated during the COVID-19 pandemic (between January 2020 and December 2022).

There were no differences in demographic parameters between COVID-19 and non-COVID-19 patients (Table 1). COVID-19 patients were longer invasively ventilated before ECMO initiation (median: 3.0 days (IQR: 1.0, 8.8) versus 2.0 (IQR: 0.4, 5.0); *p*: 0.005), and they also spent more total time on invasive mechanical ventilation (median 31.8 (IQR: 22.8; 48.4) versus 27.4 (IQR: 13.0; 49.0); *p*: 0.022). Significantly more COVID-19 patients were ventilated in prone-position prior to ECMO (71% versus 30%, *p*: 0.001) and after ECMO initiation (94% versus 46%, *p*: 0.001). Of note, pre-existing comorbidities were more frequent in non-COVID-19 patients, as documented by a higher CCI (mean 1.9 (SD 2.0) versus 0.8 (SD: 1.3); *p*: <0.001) (see Additional File 1). Also, more non-COVID-19 patients were immunosuppressed upon the initiation of ECMO (29.3% versus 2.5%), as defined by Rilinger et al. [18]. Acutely started steroid therapies, such as hydrocortisone for sepsis or dexamethasone for COVID-19, were not included in our definition.

Also, non-COVID-19 patients were in a more critical condition when ECMO was initiated: 32% were on CKRT (versus 15%, *p*: <0.001), cardiopulmonary resuscitation (CPR) had been performed in 13% (versus 3%, *p*: 0.002) and the SOFA score upon admission was higher (median 9.0 (IQR: 7.0, 11.0) versus 8.0 (IQR: 6.0, 9.0); *p*: <0.001). Twenty-four hours after admission, the SAPS-II was also higher in non-COVID-19 patients (median 47.0 (IQR: 38.0, 55.0) versus 42.0 (IQR: 38.0, 49.0); *p*: 0.007), and so were the TISS-10 values (median 28.0 (IQR: 23.0, 34.0) versus 27.0 (IQR: 22.0, 31.0); *p*: 0.009). In non-COVID-19 patients, the most frequent etiologies of respiratory failure were as follows: not-further-specified respiratory diagnoses (37%), bacterial pneumonias (22%) and viral pneumonias (18%) not caused by severe acute respiratory syndrome-coronavirus-2 (SARS-CoV2).

### 3.2. Overall Outcome

Overall survival to discharge was 41% (Table 2). Despite significantly higher RESP scores in COVID-19 patients (median 1.0 (IQR: −1.0, 2.0) versus −1.0 (IQR: −3.0, 2.0); *p*: <0.001), survival to discharge was lower (31% versus 45%, *p*: 0.007).

Long-term survival was higher in non-COVID-19 patients (Figure 2). In non-survivors, the time between ECMO initiation and death was longer for non-COVID-19 patients (median 40.7 days (IQR: 11.9, 551.5) versus 28.1 days (IQR: 18.8, 50.3); *p*: 0.029) (Table 2). COVID-19 patients spent more days on ECMO (median 20.3 (IQR: 13.8, 28.5) versus 10.0 (IQR: 6.0, 16.6); *p*: <0.001), and they were less frequently weaned off ECMO (36% versus 57%, *p*: <0.001). There were no significant differences in the ICUL LoS and in the hospital LoS.

### 3.3. Incidence of IFD and Impact on Survival of COVID-19 Patients

IFD was detected in 6% of all patients on V-V ECMO (Table 3). It occurred significantly more frequently in COVID-19 patients (11% versus 4.8%, *p*: 0.021). The OR for IFD was 2.46 for COVID-19 patients (CI 1.12–5.28; *p*: 0.021) in the univariable binary logistic regression analysis. We also performed a multivariable analysis, including SOFA score upon admission, age, gender, height, weight, duration of IMV, duration of ECMO prior to IFD (if any) and COVID-19 status as independent variables. In this analysis, IFD was associated with COVID-19 (OR 4.30; CI 1.72–10.85; *p*: 0.002) and with the duration of IMV (OR 1.015; CI 1.00–1.027; *p*: 0.012). Neither the time on ECMO before IFD (OR 0.963; CI 0.92–1.00; *p*: 0.08) nor any of the other remaining variables were significantly associated with IFD.

These results were confirmed when COVID-19 patients were matched to non-COVID-19 patients (see Methods). In the PSM cohort, the OR of COVID-19 patients for IFD was 4.74 (CI: 1.48–21.11; *p*: 0.017) in the univariable regression analysis.

Candidemia accounted for all cases of IFD in the non-COVID-19 patients, and for 84.6% of IFD in the COVID-19 patients (Table 3). C. albicans represented the most frequently found species in both of the cohorts. Two cases of pulmonary aspergillosis occurred in the COVID-19 patients. The COVID-19 patients had been ventilated longer before an IFD was detected (median 22.0 days (IQR: 17.0, 28.0) versus 14.5 days (IQR: 4.8, 22.8); *p*: 0.04). Albeit not reaching statistical significance, there was also a trend toward a longer duration of ECMO before the detection of IFD. In this cohort, the first IFD was detected after 203 patient days on ECMO. In the non-COVID-19 patients, the first IFD was detected after 247 patient days on ECMO. These patients also cleared IFDs more frequently than the COVID-19 patients (81.3% versus 38.5%, *p*: 0.027).

Overall, the detection of IFD was associated with mortality (Table 3): 42.8% (181/423) of patients without IFD were discharged alive, whereas 13.8% (4/29; *p*: 0.002) of patients with IFD survived. A total of 45.6% (145/318) of the non-COVID-19 patients without IFD survived to discharge, whereas only 25% (4/16) with IFD were discharged alive. The time of survival was significantly shorter in the non-COVID-19 patients with IFD (Figure 3a). In the COVID-19 patients (Figure 3b), 34.3% (36/105) without IFD survived to discharge, but all of the patients with IFD died. Among all of the patients with IFD, the impact of COVID-19 status on survival was not significant (Table 3).

## 4. Discussion

In this single-center, retrospective study in patients on V-V ECMO for respiratory failure, we observed an IFD incidence of 4.8% in non-COVID-19 patients and 11.0% in COVID-19 patients (Table 3). Utilizing logistic regression analysis and PSM, we identified COVID-19 as an independent risk factor for IFD. The survival of COVID-19 patients was lower, and the detection of IFD was associated with a further increase in mortality (Figure 3 and Table 3). IFD also decreased the time of survival in non-COVID-19 patients (Figure 3). A total of 93.1% of IFD represented candidemia, and we only observed two cases of possible IA in COVID-19 patients. Of note, 242 non-COVID-19 patients were treated before the pandemic, and 92 non-COVID-19 patients were treated during the COVID-19 pandemic.

The survival rate of the non-COVID-19 patients in our study is in line with other observations: overall survival was 48% in a mixed population of patients in the ELSO registry [1]. For COVID-19 patients, we observed a survival to discharge rate of 31% (Table 2). Recent studies have reported a survival rate of 27–32% for COVID-19 patients on ECMO in Germany [23,24]. In a current Italian study, the mortality rate of COVID-19 patients on ECMO even exceeded 80% [14]. For greater Paris, France, 90-day-survival was reported with 46% [25,26]. In comparison, the COVID-19 patients in our study were older (median age 57.4 years versus 52 years) and they had higher SAPS-II values (SAPS-II 42 versus 40) [26]. In summary, our observed survival rates are in line with other reports.

In non-COVID-19 patients on ECMO, the incidence of candidemia ranges from 1.2% [1] to 9% (13/145) [27]. In our study, the incidence of candidemia was 4.8%, which is similar to approx. 6% as described by Aubron et al. [28]. Survival to discharge was 25%, which is similar to that reported by some other studies [29], but lower than that reported by the ELSO registry (35.9%) [1]. Of note, the incidence of candidemia is 3.3% in the general ICU population (non-COVID-19 patients) with an LoS of seven days or more, with 43.4% survival [30].

In our study, the incidence of IFD was more than two-fold higher (11%) in COVID-19 patients (Table 3). Indeed, COVID-19 was an independent risk factor for IFD in our patients on V-V ECMO, which had not been investigated before. In our study, the incidence of candidemia was 9.3% in COVID-19 patients on ECMO. Others recently reported a similarly high incidence in critically ill COVID-19 patients on and off ECMO, ranging from 5.1% to 14.4% [12,31,32,33,34,35]. Hence, the term COVID-19-associated candidiasis (CAC) was coined [36].

Similarly to CAC, the term CAPA (COVID-19-associated pulmonary aspergillosis) reflects the perceived high incidence of pulmonary aspergillosis in COVID-19 patients. In ventilated patients, some recent studies have suggested a CAPA incidence of 14.8% [35,37] to 37.5% [38], while others have reported an incidence between 2% [39] and 4.8% [40] for COVID-19-associated invasive mold disease. Of note, case definitions varied between studies. Here, we observed an incidence of 1.7% for IA in our patient cohort, as defined by positive galactomannan assays with much additional cultural evidence.

Overall, the incidence of IFD in COVID-19 patients is high. The reasons for this remain elusive, but immunosuppression and ICU-LoS are well-recognized risk factors [3,41,42]. At the beginning of ECMO, more non-COVID-19 patients than COVID-19 patients were receiving continuous, permanent immunosuppressive therapy, which did not include steroids for sepsis. However, immunosuppression is part of therapy in COVID-19, with the addition of dexamethasone as the standard treatment for ventilated patients [43]. While some studies have demonstrated a correlation between corticosteroid use and candidemia in COVID-19 patients [44], others have not [10,37,45]. In Germany, dexamethasone (6 mg intravenously, once per day for ten days) was added to the guideline recommendations in July 2020, with the preliminary report of the corresponding RECOVERY subtrial [46]. We, however, did not detect an increase in the incidence of IFD with this change in treatment recommendations (see Additional File 2 (phase classification according to Schilling et al. [47])), arguing against dexamethasone as the cause of IFD in COVID-19 patients.

Other data suggest that SARS-CoV2 impairs T cell function [48]. COVID-19 might also induce the breakdown of mucosal barriers, such as in the lung or intestines. It is speculative whether this explains the higher incidence of candidemia in COVID-19 patients on ECMO, compared to COVID-19 patients not on ECMO, as reported by Alessandri et al. [14]. We believe that it is not ECMO itself that is causing candidemia, as in our multivariable regression analysis, time on ECMO before IFD was not associated with IFD. Furthermore, in non-COVID-19 patients on ECMO in our institution, the incidence of candidemia was only slightly higher than in the general ICU population (4.8% versus 3.3%) [30]. Of note, we observed a much lower incidence of candidemia than that reported by Alessandri et al. (9.32% versus 35.6%) [14].

We observed no significant differences in baseline characteristics between non-COVID-19 patients and COVID-19 patients. In general, patients did not present classical risk factors for IFD, such as hematological malignancies or defects in the immune system (see Appendix A). However, there were significant differences in some treatment factors: The total time on IMV was longer in our COVID-19 patients (Table 3), possibly contributing to the higher incidence of IFD [1,8,9,49,50]. In our study, COVID-19 patients also spent a longer time on ECMO (Table 2), another known risk factor for infections [27,28,29,51,52]. There was also a trend toward a longer time on ECMO before IFD (Table 3). Hence, we included the time on IMV and the duration of IFD before ECMO in our multivariable regression analysis to adjust for these confounders. Still, the OR for IFD was 4.3 in the COVID-19 patients. Further, the COVID-19 patients were ventilated in prone-position more often (Table 1), which could theoretically increase the risk for catheter infections, as central-venous catheters and ECMO cannulas may become exposed to bodily secretions. On the other hand, the COVID-19 patients were in a less critical condition, while severity of illness was a risk factor for candidemia in earlier studies [53]. In addition, less COVID-19 patients had CKRT, and renal failure has been identified as another risk factor for candidemia [1,8,9,54,55,56,57,58]. Other supportive treatments, such as parenteral nutrition and broad-spectrum antibiotic therapy, were highly standardized in our institution, as outlined before [2]. Hence, we argue against these factors as the cause of the higher incidence of IFD in COVID-19 patients.

Finally, the COVID-19 pandemic put significant strain on healthcare systems. It is conceivable that overwhelming numbers of patients led to reductions in the staff-per-patient ratio, consequently lowering hygiene standards. This might have promoted IFD during the pandemic in some regions. This argument is also supported by a higher incidence of candidemia in non-COVID-19 patients during the pandemic [59]. In the same study, the incidence of candidemia was still higher in COVID-19 patients. We argue against the breakdown of hygiene standards as an explanation for our results: For ECMO patients in our institution, the regular staff-per-patient ratio was continued throughout the pandemic. Furthermore, the number of COVID-19 patients on ECMO changed significantly over the course of the pandemic, whereas the incidence of IFD in these patients did not. Hence, we argue that a breakdown in hygiene did not cause the high incidence of IFD in COVID-19 patients compared to non-COVID-19 patients.

Taken together, our study does not allow for firm conclusions on the reasons for the increased incidence of IFD in COVID-19 patients. However, it is conceivable that these patients contracted IFD as a consequence of a prolonged disease, with long ECMO runs.

Equally important as the reason for the high incidence of IFD in COVID-19 patients is its impact on survival: in a large French cohort of ventilated COVID-19 patients not on ECMO, univariable analysis revealed an association between death and candidemia. However, this was not confirmed via multivariable analysis [37]. Other studies have observed higher mortality in ventilated COVID-19 patients with IFD [10,34,35]. For non-COVID-19 patients on ECMO, data from the ELSO registry show that candidemia and IA are associated with increased mortality [1]. In COVID-19 patients on ECMO, a recent British study demonstrated an association of IA with mortality. Here, we demonstrate for the first time that CAC is also associated with a dire prognosis in patients on ECMO (survival: 0%, sic!). We emphasize that, in our study, IFDs were associated with a bad prognosis. It remains unknown as to whether they represent the cause or consequence.

Of note, neither the prognostic impact of COVID-19 nor the relevance of IFD are reflected by the RESP score. Remarkably, the COVID-19 patients in our study had better RESP scores but showed lower survival.

Our study is limited by its retrospective, monocentric nature, limiting the number of cases. Furthermore, the diagnosis of definitive IFD according to current guidelines [3] was not possible, due to the impracticality of frequent CT-imaging and tissue biopsies in our patient population. Screening for IFD was performed at the time of ECMO initiation (blood cultures and bronchoalveolar lavage). Further tests for IFD were performed at the discretion of the treating physician, with every new insertion of central lines and with every bronchoscopy. Hence, the true incidence of IFD is probably underestimated. Furthermore, our regression analyses did not include all treatment factors potentially predisposing patients to IFD.

To the best of our knowledge, our study is the first demonstrating COVID-19 to be an independent risk factor for IFD in patients on V-V ECMO for respiratory failure. For the first time, we provide evidence that IFD in general and candidemia in particular are associated with a striking increase in mortality in COVID-19 patients on ECMO.

## 5. Conclusions

In patients on V-V ECMO, we demonstrate a higher incidence of IFD and candidemia in COVID-19 patients. IFD in general and candidemia by itself are associated with decreased survival in this patient population. Future studies should determine when (prophylaxis versus proven disease) and how (the relevance of biofilms in ECMO patients) these infectious complications should be treated.

## Figures and Tables

**Figure 1 jof-09-00751-f001:**
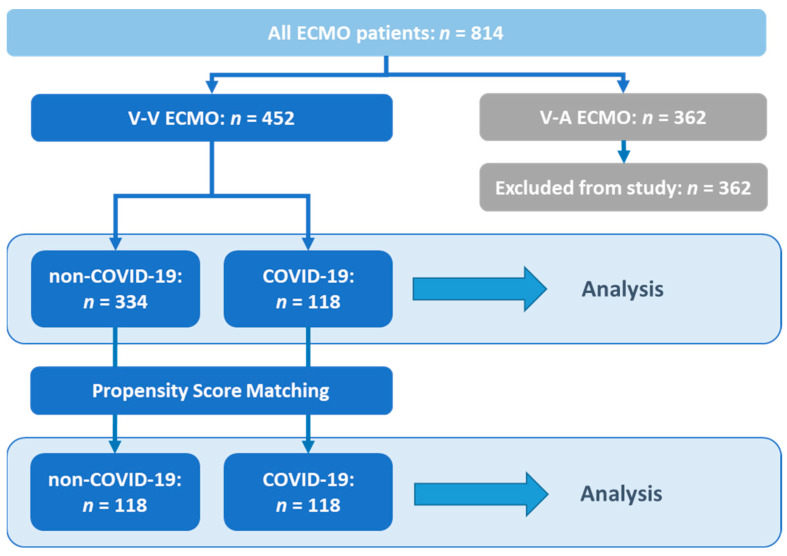
Inclusion process for the selected patient cohort. ECMO: extracorporeal membrane oxygenation. V-A: veno-arterial. V-V: veno-venous.

**Figure 2 jof-09-00751-f002:**
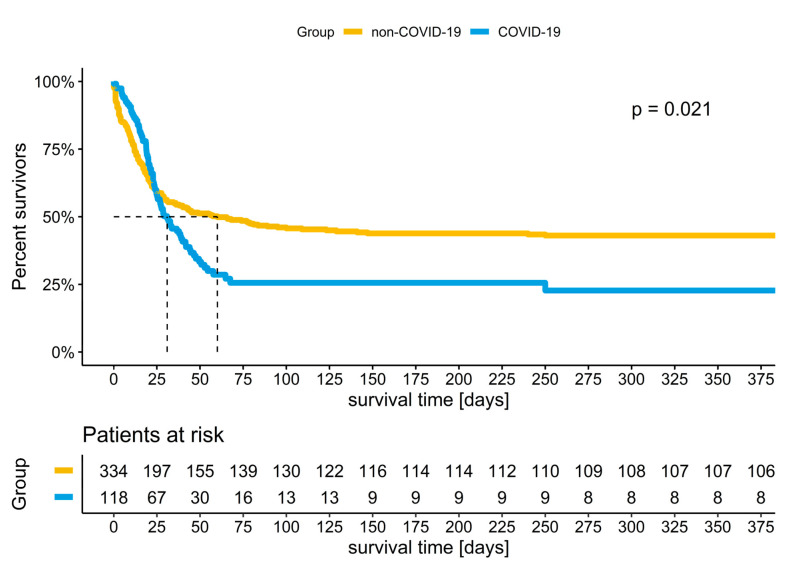
Kaplan–Meier analysis: survival in ECMO patients with and without COVID-19.

**Figure 3 jof-09-00751-f003:**
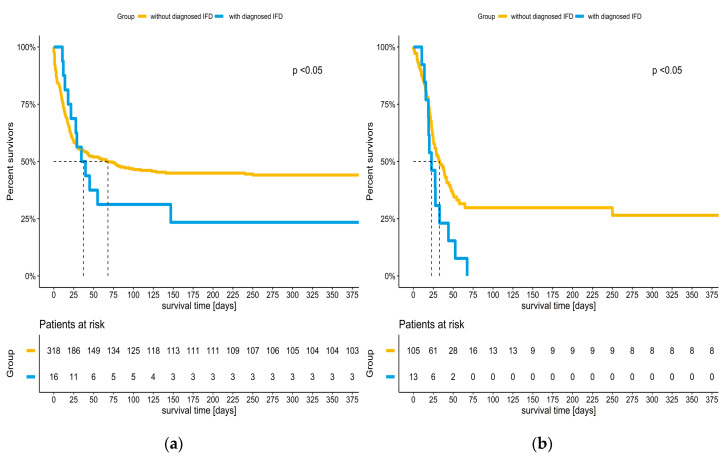
Kaplan–Meier analysis: survival in ECMO patients with and without IFD, stratified by COVID-19 status: (**a**) survival in non-COVID-19 patients; (**b**) survival in COVID-19 patients.

**Table 1 jof-09-00751-t001:** Demographics and further characterization of the study population.

COVID-19 Status (*n*)	Total (452)	Non-COVID-19 (334)	COVID-19 (118)	*p*
**Demographics**				
Age (years) upon ICU admission (median, [IQR])	55.9 [47.0; 64.0]	54.9 [45.7; 64.5]	57.4 [49.8; 62.2]	0.519
Height (cm (median [IQR]))	175.0 [168; 180.0]	175.0 [167.0; 180.0]	175.0 [170.0; 180.0]	0.192
Weight (kg (median [IQR]))	90.0 [80.0; 110.0]	90.0 [76.2; 110.0]	90.0 [85.0; 104.2]	0.064
BMI (kg/m^2^ (median [IQR]))	29.2 [26.1; 35.1]	29.0 [25.0; 34.8]	29.4 [27.6; 35.1]	0.051
Gender (female (%))	150 (33)	115 (34)	35 (30)	0.344
**Ventilation**				
Days of IMV prior to ECMO (median [IQR])	2.0 [0.9; 6.0]	2.0 [0.4; 5.0]	3.0 [1.0; 8.8]	0.005
Total days on IMV (median [IQR])	28.9 [15.0; 49.0]	27.4 [13.0; 49.0]	31.8 [22.8; 48.4]	0.022
Tracheotomy at any time (yes (%))	201 (45)	148 (44)	53 (45)	0.873
Proning prior to ECMO (yes (%))	183 (40)	99 (30)	84 (71)	<0.001
Proning on ECMO (yes (%))	265 (59)	154 (46)	111 (94)	<0.001
**Other Organ Dysfunction**				
SAPS-II 24 h after ECMO initiation (median [IQR])	45.5 [38.0; 54.2]	47.0 [38.0; 55.0]	42.0 [38.0; 49.0]	0.007
SAPS-II upon discharge (median [IQR])	46.5 [31.0; 60.0]	45.0 [30.0; 59.5]	48.0 [35.0; 60.5]	0.267
TISS-10 24 h after ECMO initiation (median [IQR])	27.0 [22.2; 33.0]	28.0 [23.0; 34.0]	27.0 [22.0; 31.0]	0.009
TISS-10 upon discharge (median [IQR])	23.5 [10.0; 31.0]	22.0 [10.0; 30.0]	27.0 [15.0; 33.0]	0.009
SOFA score (median [IQR])	9.0 [7.0; 10.0]	9.0 [7.0; 11.0]	8.0 [6.0; 9.0]	<0.001
CKRT prior to ECMO (yes (%))	125 (28)	107 (32)	18 (15)	<0.001
CPR prior to ECMO (yes (%))	45 (10)	42 (13)	3 (3)	0.002
CCI (mean (SD))	1.6 (1.9)	1.9 (2.0)	0.8 (1.3)	<0.001
**Etiology of Respiratory Failure**				
Not specified	1 (0.2)	1 (0.3)		
Aspiration pneumonitis (*n* (%))	39 (9)	39 (12)		
Asthma (*n* (%))	4 (1)	4 (1)		
Bacterial pneumonia (*n* (%))	72 (16)	72 (22)		
Non-respiratory and chronic respiratory diagnoses (*n* (%))	25 (6)	25 (7)		
Other acute respiratory diagnosis (*n* (%))	125 (28)	125 (37)		
Trauma/burn (*n* (%))	8 (2)	8 (2)		
Viral pneumonia (*n* (%))	178 (39)	60 (18)	118 (100)	

Legend: BMI: body mass index. CCI: Charlson Comorbidity Index. CPR: cardio-pulmonary resuscitation. CKRT: continuous kidney replacement therapy. ECMO: extracorporeal membrane oxygenation. IMV: invasive mechanical ventilation. SAPS-II: Simplified Acute Physiology Score II. SOFA: Sequential Organ Failure Assessment. TISS-10: Therapeutic Intervention Scoring System-10.

**Table 2 jof-09-00751-t002:** Outcome parameters.

COVID-19 Status (*n*)	Total (452)	Non-COVID-19 (334)	COVID-19 (118)	*p*
RESP score (median [IQR])	−1.0 [−3.0; 2.0]	−1.0 [−3.0; 2.0]	1.0 [−1.0; 2.0]	<0.001
Time on ECMO (days (median [IQR]))	12.0 [7.6; 20.0]	10.0 [6.0; 16.6]	20.3 [13.8; 28.5]	<0.001
ECMO weaning successful (yes (%))	234 (52)	191 (57)	43 (36)	<0.001
Survival to discharge (yes (%))	185 (41)	149 (45)	36 (31)	0.007
Days from ECMO initiation to death in non-survivors (median [IQR])	34.0 [13.9; 382.8]	40.7 [11.9; 551.5]	28.1 [18.8; 50.3]	0.029
ICU: length of stay (days (median [IQR]))	25.9 [14.0; 47.9]	24.9 [12.9; 48.4]	28.0 [18.4; 46.5]	0.278
Hospital: length of stay (days (median [IQR]))	28.1 [15.0; 53.5]	28.1 [14.1; 57.3]	28.0 [18.4; 46.5]	0.849

Legend: ECMO: extracorporeal membrane oxygenation. ICU: intensive care unit. RESP: respiratory ECMO survival prediction.

**Table 3 jof-09-00751-t003:** Invasive fungal disease during ECMO.

COVID-19 Status (*n*)	Total (452)	Non-COVID-19 (334)	COVID-19 (118)	*p*
IFD during ECMO (yes (% of all patients))	29 (6.4) ^#^	16 (4.8)	13 (11.0)	0.018
**Type of IFD (*n* (% of IFDs in cohort))**				
** Candidemia**	**27 (93.1)**	**16 (100.0)**	**11 (84.6)**	
C. albicans	19 (65.5) **^#^**	13 (81.3)	6 (46.2) ^#^	
C. glabrata	3 (10.3)	1 (6.25)	2 (15.4)	
C. krusei	2 (6.9)	1 (6.25)	1 (7.7)	
C. parapsilosis	2 (6.9)	1 (6.25)	1 (7.7)	
C. kefyr	1 (3.4)		1 (7.7)	
C. dubliniensis	1 (3.4) ^#^		1 (7.7) **^#^**	
** Invasive aspergillosis**	**2 (0.4)**		**2 (15.4)**	
Asp. fumigatus	1 (0.2)		1 (7.7)	
Asp. terreus	1 (0.2)		1 (7.7)	
Time on IMV before IFD (days (median [IQR]))	17.0 [9.0; 27.0]	14.5 [4.8; 22.8]	22.0 [17.0; 28.0]	0.04
Time on ECMO before IFD (days (median [IQR]))	12.0 [0.0; 18.0]	8.5 [0.0; 14.8]	13.0 [6.0; 23.0]	0.084
Clearance of IFD (yes (% of all IFD))	18 (62.1)	13 (81.3)	5 (38.5)	0.027
Time to clearance (days (median [IQR]))	3.5 [2.0; 5.0]	4.0 [3.0; 5.0]	2 [2.0; 2.0]	
Survival to discharge (yes (% of patients with IFD))	4 (13.8)	4 (25.0)	0 (0.0)	0.107

Legend: IFD: invasive fungal disease. ECMO: extracorporeal membrane oxygenation. C.: Candida. Asp.: Aspergillus. IMV: invasive mechanical ventilation. #: Double infection with C. albicans and C. dubliniensis in one candidemic patient.

## Data Availability

The datasets analyzed for the current study are available from the corresponding author upon reasonable request.

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
