# Peer review of "COVID-19 Is an Independent Risk Factor for Detrimental Invasive Fungal Disease in Patients on Veno-Venous Extracorporeal Membrane Oxygenation: A Retrospective Study"

_jof, 2023, doi:10.3390/jof9070751_

Round 1
Reviewer 1 Report
This is an interesting manuscript considering the frequence of invasive fungal disease in patients using extracorporeal membrane oxygenation in the presence or not of COVID-19, observed in a single center. It is well-written and conclusions are pertinent. One limitation and fail is the lack of an improved diagnosis of fungal infections that could demonstrate other agents.
Author Response
Reply:
We thank the reviewer for his time and insight. We agree with the reviewer that a more sophisticated diagnostic approach would have been helpful. In critically ill patients, however, diagnosis of invasive fungal disease (IFD) remains challenging, as reviewed in [1,2]. Up to date, neither PCR-based detection of fungal DNA, nor the detection of other fungal components (e.g., beta-D-glucan) have improved treatment success ([3]; also see “CandiSep”, amongst authors C. Putensen, [4]). In our retrospective, single-center analysis, we utilized available real-life data. Hence, IFD were defined as outlined in our manuscript. We believe that the observed frequency of IFD was probably lower than the true incidence, due to the diagnostic limitations pointed out also by the reviewer. We emphasize this in the second last paragraph of the discussion (ll. 316).
- Martin-Loeches, I.; Antonelli, M.; Cuenca-Estrella, M.; Dimopoulos, G.; Einav, S.; De Waele, J.J.; Garnacho-Montero, J.; Kanj, S.S.; Machado, F.R.; Montravers, P.; et al. ESICM/ESCMID Task Force on Practical Management of Invasive Candidiasis in Critically Ill Patients. Intensive Care Med 2019, 45, 789–805, doi:10.1007/s00134-019-05599-w.
- Logan, C.; Martin-Loeches, I.; Bicanic, T. Invasive Candidiasis in Critical Care: Challenges and Future Directions. Intensive Care Med 2020, 46, 2001–2014, doi:10.1007/s00134-020-06240-x.
- Pappas, P.G.; Kauffman, C.A.; Andes, D.R.; Clancy, C.J.; Marr, K.A.; Ostrosky-Zeichner, L.; Reboli, A.C.; Schuster, M.G.; Vazquez, J.A.; Walsh, T.J.; et al. Clinical Practice Guideline for the Management of Candidiasis: 2016 Update by the Infectious Diseases Society of America. Clinical Infectious Diseases 2016, 62, e1–e50, doi:10.1093/cid/civ933.
- Bloos, F.; Held, J.; Kluge, S.; Simon, P.; Kogelmann, K.; de Heer, G.; Kuhn, S.-O.; Jarczak, D.; Motsch, J.; Hempel, G.; et al. (1 → 3)-β-D-Glucan-Guided Antifungal Therapy in Adults with Sepsis: The CandiSep Randomized Clinical Trial. Intensive Care Med 2022, 48, 865–875, doi:10.1007/s00134-022-06733-x.
Reviewer 2 Report
This single center retrospective study invesigated the relevance of Coronavirus-Disease-19 (COVID-19) for the occurrence of invasive fungal disease (IFD) in patients on veno-venous (V-V) ECMO for respiratory failure. They found that COVID-19 was an important risk factor of IFD in patients with respiratroy failure requiring V-V ECMO. Although this topic is interesting, I have several concerns.
1. The confounding effect of comorbidities, such as immunocompromised status, diabetes, cancer, transplant, were not taken into analysis.
2. Systematic corticosteroid is the standard treatment for severe COVID-19, but also the risk factor of IFD. However, the effect of systemic corticosteroid was not assessed.
3. The use of TPN and intra-abdominal pathology were the common risk factor for invasive candidiasis, but was not assessed in this study.
Author Response
Reviewer 2: This single center retrospective study investigated the relevance of Coronavirus-Disease-19 (COVID-19) for the occurrence of invasive fungal disease (IFD) in patients on veno-venous (V-V) ECMO for respiratory failure. They found that COVID-19 was an important risk factor of IFD in patients with respiratroy failure requiring V-V ECMO. Although this topic is interesting, I have several concerns.
Comment 1:
- The confounding effect of comorbidities, such as immunocompromised status, diabetes, cancer, transplant, were not taken into analysis.
Reply:
We thank the reviewer for his valuable, thoughtful comments. The reviewer points to the relevance of comorbidities for the contraction of IFD. We agree that these comorbidities represent important potential confounders. For this reason, we already included some of these aspects (diabetes, cancer) in the manuscript as Additional File 1 (individual items of the Charlson Comorbidity Index). In the “Methods” section, we now added two sentences on the definition of immunocompromise (ll. 73). We also included two sentences in the “Results” section (ll. 132) and we further elaborate on our findings in the discussion (ll. 245): Of note, the incidence of conditions predisposing for IFD was higher in non-COVID-19 patients. Still, COVID-19 patients contracted IFD more often. We could not include those conditions as independent variables in our regression analysis, because the relatively small number of events (i.e., IFD cases) prohibits adding more variables to the analysis. Overfitting would be likely, confidence in findings would decrease [5].
Comment 2:
Systematic corticosteroid is the standard treatment for severe COVID-19, but also the risk factor of IFD. However, the effect of systemic corticosteroid was not assessed.
Reply:
Please refer to our reply to comment 3.
Comment 3:
The use of TPN and intra-abdominal pathology were the common risk factor for invasive candidiasis, but was not assessed in this study.
Reply to Comment 2 / Comment 3:
The reviewer raises two important concerns: a) Patient factors and b) treatment factors, such as steroids and TPN started with / after initiation of ECMO might predispose for IFD.
In our manuscript, many permanent, continuous patient and treatment factors were analyzed, such as diabetes, kidney disease, renal replacement therapy etc. The results of our analysis can be found in Additional File 1 (Charlson Comorbidity Index) and in the main text of our manuscript: The RESP score (which includes the item “immunosuppression”) is presented in ll. 152 (first paragraph of “3.2 Overall Outcome”) and in table 2, it is further discussed in ll. 313. Continuous immunosuppression is also investigated separately, as defined in the “Methods” section and further described and discussed in ll. 132 (“3.1 Study Population and Patient Characteristics”) and ll. 245.
Steroid treatment initiated with / on ECMO was not included in our analysis. However, in COVID-19 patients, we observed the highest incidence of IFD during the first wave of the pandemic, that is before the addition of dexamethasone to treatment guidelines. This argues against dexamethasone as the most relevant factor for contraction of IFD. We discuss these aspects in detail (ll. 249 and Additional File 2). We also state that systemic steroids for sepsis on ECMO (or exacerbation of COPD) were also not included in our analysis (ll. 133 and ll. 247).
Regarding TPN, we followed current local guidelines [6], which favored enteral nutrition over TPN throughout the entire study period. Hence, all patients were treated according to the same standard. However, we cannot make a definitive statement whether COVID-19 patients received TPN more often compared to non-COVID-19 patients. The same applies to intra-abdominal pathologies, such as intestinal obstruction. If intra-abdominal pathologies occurred more often in COVID-19 patients, it would be conceivable that this is due to the disease (COVID-19) itself, either directly or indirectly. Hence, our conclusion that COVID-19 is a risk factor for IFD remained true. This would also apply to the aforementioned treatment-factors, such as systemic dexamethasone and TPN.
Still, these factors might represent co-dependent variables. We refrained from including them in our regression analysis, as the sample size is relatively small (29 patients with IFD as dependent variable). This precludes the addition of further variables to the analysis (please also refer to our reply to comment 1). We added another sentence commenting further on the limitations of our study regarding this aspect (ll. 322, last sentence of the second-last paragraph of “Discussion”).
If reviewer 2 deems additional analyses compulsory, we would kindly ask for a precise definition of a) dose and duration of steroids started with / after initiation of ECMO deemed immunosuppressive; b) components, dose and duration of TPN to be considered; and c) intra-abdominal pathologies to be considered. Again, in our opinion, inclusion of these parameters in the regression analysis is precluded by the sample size. Also, such an analysis would need more time than what is granted by JoF for the revision.
- Peduzzi, P.; Concato, J.; Kemper, E.; Holford, T.R.; Feinstein, A.R. A Simulation Study of the Number of Events per Variable in Logistic Regression Analysis. J Clin Epidemiol 1996, 49, 1373–1379, doi:10.1016/s0895-4356(96)00236-3.
- Elke, G.; Hartl, W.H.; Kreymann, K.G.; Adolph, M.; Felbinger, T.W.; Graf, T.; De Heer, G.; Heller, A.R.; Kampa, U.; Mayer, K.; et al. Clinical Nutrition in Critical Care Medicine – Guideline of the German Society for Nutritional Medicine (DGEM). Clinical Nutrition ESPEN 2019, 33, 220–275, doi:10.1016/j.clnesp.2019.05.002.
Reviewer 3 Report
|
The authors of the manuscript entitled, COVID-19 Is an Independent Risk Factor for Detrimental Invasive Fungal Disease in Patients on Veno-Venous Extracorporeal Membrane Oxygenation: A Retrospective Study, present data for a total of 452 patients. |
/
Author Response
Reviewer 3:
The authors of the manuscript entitled, COVID-19 Is an Independent Risk Factor for Detrimental Invasive Fungal Disease in Patients on Veno-Venous Extracorporeal Membrane Oxygenation: A Retrospective Study, present data for a total of 452 patients.
The authors are providing good information regarding a small number of systemic fungal infections (29/452 patients) on veno-venous extracorporeal membrane oxygenation. Among the 452 patients, there were 118 Covid and 334 non Covid patients. But some changes and clarifications could improve the MS.
Comment 1:
There is a need to clarify since the Abstract, that "covid -"are non-c0vid patients that were diagnosed before the pandemia (or before 2019). On the other hand, Covid + are the patients diagnosed with this infection during the pandemia. Again, this needs to be clarified since the abstract, text and graphical items. Please, make the results are easy to interpret. (covid and non-covid will be good terms).
Reply:
We thank the reviewer for his valuable comments. As suggested by the reviewer, we changed the designation of patients from „COVID19+“ to „COVID-19 patients“ and from „COVID19-„ to „non-COVID-19 patients“ throughout the entire manuscript. We hope that this makes it easier to read and follow.
The reviewer also correctly states that „COVID19+“ refers to patients with SARS-CoV2-induced pneumonia treated throughout the pandemic (at our institution, first cases were treated in 03/2020). „COVID19- patients“ (now non-COVID-19 patients), however, refers to all SARS-CoV2-negative patients treated at our institution between 2013 and 2022. Of these patients, 92 were treated during the pandemic – still, they were SARS-CoV2-negative. In other words, of 210 patients treated during the pandemic, 118 were SARS-CoV2-positive and 92 were SARS-CoV2-negative. We added an additional sentence in the first paragraph of the “Results” section (ll. 116) and in the discussion (last sentence of the first paragraph, ll. 214). However, we chose to stick with the term COVID-19 and non-COVID-19, as the cohorts were defined by COVID-19-status and not by the time of ECMO-treatment.
Comment 2:
The paper could also be shortened, especially the discussion.
Reply:
We tried to be as precise and concise as possible. In our opinion, information would be lost if the manuscript was shortened any further. We already included much information that was considered essential by reviewer 2 in additional files (and not in the main manuscript). Indeed, we even had to add additional information to reply to essential comments of reviewer 2 and reviewer 3 (such as the distribution of non-COVID-19 patients over time). If reviewer 3 insists on shortening the manuscript, we kindly ask for further clarification which sections / which information should be dropped out.
Comment 3:
It's surprising that no diabetic patients were in the group for potential mucormycosis infections? Some more specific comments are below.
Reply:
We kindly refer the reviewer to “Additional file 1” listing all items of the Charlson Comorbidity Index, including details on diabetes in COVID-19 and non-COVID-19 patients. We did not identify cases of mucormycosis in our study population.
Comment 4:
Line 19, the COVID pandemia officially began in 2019, so the authors need to clarify that the "January 2013-2019" were pre- pandemic or non-covid cases: e.g., Figure and Table 1.
Reply:
We refer to our reply to the reviewer’s comment 1. We added clarifying information as outlined above.
Comment 5:
Again, "length of survival": Line 26-on, avoid starting a sentence with a number. More importantly, need to clearly label the columns in the tables and figures as covid patients from the others (non-covid, especially before 2019) in a crystal clear way. An as mentioned above, perhaps covid and non-covid could be clearer than the symbols - and +, which are too small anyway y.
Reply:
We changed the sentence in line 26 (abstract), so not to begin with a number. We also re-labelled the patient cohorts throughout the entire manuscript, including all tables and figures (please refer to comment 1).
Comment 6:
Lines 512-72. It would be nice to have the totals instead of only the percentages; It is evident that having covid was the most prominent death factor? That factor was not mentioned until line 91-on.
Reply:
We kindly ask the reviewer to clarify the precise position in the manuscript he is referring to, as the discussion ends with line 322. We would be glad to include absolute numbers where required, as long as the manuscript stays readable. In addition, all numbers are also included in the tables.
The reviewer correctly states that COVID-19 was the most outcome-relevant factor. We agree and we discuss this in the appropriate sections (Results: Overall Outcome. Discussion: Second paragraph). Since the incidence of IFD, and not survival, was at the center of our study, we believe that the presentation and discussion in these sections is appropriate.
Round 2
Reviewer 2 Report
The authors response well, so I have no more comment.
Author Response
We thank the reviewer for his/her time and effort.
Reviewer 3 Report
I see, as noted, that the article was corrected, except that the date was left alone in the abstract. later on, it could give the wrong impression that we had covid pates since 2013. This should be important since the number of covid patients was smaller.
/
Author Response
We thank the reviewer for mentioning this. We clarified the abstract and highlighted the revised passage (page 1 line 19-20). The sentence is now including the specific timeframe for the COVID-19 patients and reads as follows: "In a retrospective analysis of all ECMO cases between January 2013 to December 2022 (2020-2022 for COVID-19 patients), active COVID-19, type, timing and duration of IFD were investigated. "
We appreciate the reviewers time and insights for our manuscript.